# UV Polymerization of Methacrylates—Preparation and Properties of Novel Copolymers

**DOI:** 10.3390/polym13101659

**Published:** 2021-05-20

**Authors:** Marta Worzakowska

**Affiliations:** Department of Polymer Chemistry, Institute of Chemical Sciences, Faculty of Chemistry, Maria Curie-Skłodowska University, Gliniana 33 Street, 20-614 Lublin, Poland; marta.worzakowska@poczta.umcs.lublin.pl; Tel./Fax: +48-81-524-22-51

**Keywords:** UV polymerization, copolymers, geranyl methacrylate, citronellyl methacrylate, methyl methacrylate, properties

## Abstract

More environmentally friendly polymeric materials for use in corrosive conditions were obtained in the process of UV polymerization of terpene methacrylate monomers: geranyl methacrylate and citronellyl methacrylate and the commercially available monomer methyl methacrylate. Selected properties (solvent resistance, chemical resistance, glass transition temperature, thermal stability, and decomposition course during heating) were evaluated. It was found that the properties of the materials directly depended on the monomer percentage and the conditioning temperatures used. An increase in the geranyl or citronellyl methacrylate monomer content in the copolymers reduced the solubility and chemical resistance of the materials post-cured at 50 °C. The samples post-cured at 120 °C were characterized by high resistance to polar and non-polar solvents and the chemical environment, regardless of the percentage composition. The glass transition temperatures for samples conditioned at 120 °C increased with increasing content of methyl methacrylate in the copolymers. The thermal stability of copolymers depended on the conditioning temperatures used. It was greater than 200 °C for most copolymers post-cured at 120 °C. The process of pyrolysis of copolymers led to the emission of geranyl methacrylate, citronellyl methacrylate, and methyl methacrylate monomers as the main pyrolysis volatiles.

## 1. Introduction

The UV polymerization process is a process of monomer polymerization in the presence of a photoinitiator. The photoinitiator absorbs UV radiation within the appropriate wavelength range (300–400 nm). Thus, it dissociates into radicals that initiate the polymerization of monomers with a defined structure [1,2]. Among the monomers, the most frequently used monomers are acrylate and methacrylate monomers, newly synthesized or commercially available [3,4,5,6,7,8,9,10].

The UV polymerization process has attracted much interest due to many advantages. This process is very fast and saves energy. It does not require the use of additional solvents. It is carried out at ambient temperature. This process can be controlled and take place where the sample is irradiated. UV polymerization also has some disadvantages, such as curing only thin layers, the need for additional lamp devices, and the need to eliminate visible light, which adversely affects the process. However, despite these drawbacks, the UV polymerization process is widely used. It allows us to obtain a wide range of polymers with predetermined or unique physico-chemical properties in a short time. UV polymerization has been widely used in the preparation of polymer-based photoactive systems that are used in the paint, coating, adhesive, and printing industries, for obtaining composite materials and optical fibers, and in microelectronics for over 30 years. It is also used in less traditional but interesting applications including laser video drives, curable dental fillings, the production of 3D objects, the production of biomaterials used as bone in tissue engineering, and obtaining photosensitive materials and microchips [11,12,13,14,15,16,17,18,19,20].

According to a literature review, the use of terpene and terpenoid compounds in the production of biomaterials, both from a sustainable development perspective and an environmental protection perspective, is widely studied. Terpenes and terpenoids are compounds of natural origin produced by plants. They are made from relatively quickly renewable sources and thus can be easily studied. The preparation and testing of new biomaterials are very important, because currently above 7% of all fossil fuels (petroleum, coal) extracted worldwide is used for the production of plastics. Consequences of the increasing mining of non-renewable materials (petroleum, coal, and gas) include increasing amounts of greenhouse gases, lower air quality, and global warming. Thus, it seems advisable to undertake research on the use of natural compounds or derivatives of natural compounds in the preparation of biopolymers, which, due to their properties, could replace synthetic polymers in many applications. Among the terpene compounds, the polymerization of β-pinene, monomers derived from pinene and limonene, and the polymerization of pinene with styrene or acrylates have been investigated [21,22,23,24,25]. Moreover, researchers have described the polymerization of citronellol oxide obtained from citronellol, leading to the production of hyperbranched polymers [26,27,28], and the graft polymerization of methacrylate ester derivatives of citronellol, geraniol, or nerol with starch in order to obtain more environmentally friendly graft materials with modified physicochemical properties [29,30,31,32].

The main purpose of this study is to obtain novel, more environmentally friendly polymeric materials, using different monomer compositions, for use in corrosive conditions and to evaluate their properties (solvent and chemical resistance, glass transition temperatures, thermal resistance, and decomposition course under heating). The influence of the structure of the obtained polymeric materials as well as post-curing temperatures on the abovementioned properties was investigated and is discussed.

## 2. Materials and Methods

### 2.1. Materials

Methyl methacrylate (MM) (≥99%), methacryloyl chloride (97%), geraniol (trans-3,7-dimethyl-2,6-octadien-1-ol, 98%), citronellol (trans-3,7-dimethyl-2,6-octadien-1-ol, 95%), and trimethylamine (≥99.5%) were purchased in Sigma-Aldrich (St. Louis, MO, USA). Irgacure 651 (2,2,-dimethoxy-1,2-diphenylethan-1-one), methanol, chloroform, hexane, toluene, butanol, and silica gel were obtained from Merck. Sodium carbonate, magnesium sulfate, sodium hydroxide, carbon tetrachloride, hydrochloric acid, magnesium sulfate, and buffer solutions (pH 5, 7, and 9) were obtained from POCh, Gliwice, Poland.

### 2.2. Synthesis of Methacrylate Monomers

Methacrylate ester monomers (geranyl methacrylate and citronellyl methacrylate) were obtained in the process of esterification of methacryloyl chloride with one of the natural terpene alcohols (geraniol or citronellol) in the presence of trimethylamine according to the procedure described in [30,31,32,33]. Terpene methacrylate esters were used as monomers in the preparation of copolymers with the commercially available methyl methacrylate (MM) monomer. The structures of the obtained terpene methacrylate monomers were confirmed and are described in [34].

### 2.3. UV-Polymerization

UV polymerization of geranyl methacrylate (GM), citronellyl methacrylate (CM), methyl methacrylate (MM), and monomer mixtures of various compositions in the presence of Irgacure 651 (3% mass) was initiated with a TL20W/05 SLV low-pressure mercury lamp (340–365 nm). The samples (circle-shaped samples with a diameter of 30 mm and a thickness of 5 mm) were irradiated for 10 min at 25 °C. After irradiation, all samples were conditioned, first at 50 °C for 5 h (series 1) and then at 120 °C for 3 h (series 2), Table 1.

### 2.4. Characterization of Copolymers

#### 2.4.1. ATR-FTIR

The ATR-FTIR spectra for methacrylate monomers, homopolymers, and copolymers were collected in the range 600–4000 cm^−1^ and at a 4 cm^−1^ resolution using 64 scans per spectrum. A FTIR Tensor 27 equipped(Bruker, Germany) with a diamond crystal produced by Bruker was applied.

#### 2.4.2. Conversion of Double Bonds

The conversion of methacrylate monomers was measured by FTIR spectroscopy. The conversion was determined by the comparison of the area of the C=C stretching vibrations band at 1633 cm^−1^ (methacrylate bonds) and at 1672 cm^−1^ (ethylenic bonds) with the area of the C=O stretching vibration band at 1716 cm^−1^. The conversion degree (DC) of the analyzed polymeric materials was calculated by the following equation:DC/% = 100 × [1 − (R_polymer_/R_monomer_)]
where R is the surface area of the C=C absorption band/surface area of the C=O absorption band.

#### 2.4.3. Solubility Tests

The solubility tests for the obtained copolymers were carried out at 25 °C with the following solvents: water, methanol, butanol, toluene, hexane, carbon tetrachloride, and chloroform. The sample mass was approximately 0.2 g. The samples were poured into a suitable solvent (10 mL) and kept in these solvents until constant mass was obtained. Then, the solvent was filtered off. The polymer sample was dried on tissue paper and weighed. The solubility tests were carried out over a period of 6 months. The solubility was determined based on the equation:Δ*mS* = (*m*_1_ − *m*_2_)/*m*_1_ × 100%
where *m*_1_—the initial mass of the sample, *m*_2_—the final mass of the sample, and Δ*mS*—the percentage change in mass.

#### 2.4.4. Chemical Resistance

The chemical resistance tests for copolymers were performed using 1M NaOH, buffer solutions with pH 5, 7, and 9, and 1M HCl. About 0.2 g of the sample was immersed in the solutions (10 mL). The sample solutions were left until constant mass was obtained. Then, the samples were filtered, carefully washed with distilled water, dried, and weighed. The percentage change in mass loss (Δ*mR*) for materials was evaluated from the equation given in [33]:Δ*mR* = (*m*_1_ − *m*_2_)/*m*_1_ × 100%
where *m*_1_—the initial mass of the sample and *m*_2_—the final mass of the sample.

#### 2.4.5. Glass Transition Temperature

The glass transition temperature (*T*_g_) of the obtained polymeric materials was evaluated with a use of a DSC 204 calorimeter (Netzsch, Selb Germany). The heating of the materials was performed in two scans from −120 °C to 120 °C with a heating rate of 10 K min^−1^. The sample mass was ca. 10 mg. The analyses were done in aluminum crucibles with a pierced lid under an argon atmosphere (a flow rate of 40 mL min^−1^). The calorimeter was calibrated according to the manufacturer’s instructions. *T*_g_ was read from the second DSC scan.

#### 2.4.6. Thermal Properties

The TG/DTG analyses for the tested materials were performed using a STA 449 Jupiter F1 instrument produced by Netzsch, Selb, Germany. Ten milligrams (10 mg) of the sample was put into an Al_2_O_3_ crucible and heated between the temperatures of 40 °C and 550 °C with a heating rate of 10 K min^−1^ in the presence of a helium atmosphere (a flow rate of 40 mL min^−1^). The initial decomposition temperatures (*T*_5%_), maximum decomposition temperatures (*T*_max_), and mass losses (Δ*m*) at each stage of the decomposition and the residual masses at 550 °C (rm) were determined.

#### 2.4.7. Simultaneous TG–FTIR Analysis

The decomposition course of the tested materials during heating in an inert atmosphere was monitored by simultaneous TG–FTIR analysis. The FTIR analyzer (FTIR TGA 585, Bruker, Mannheim, Germany) was connected on-line to a STA instrument through a Teflon tube of 2 mm in diameter heated to 200 °C. The FTIR spectra of volatiles were collected from 600 cm^−1^ to 4000 cm^−1^ at a 4 cm^−1^ resolution.

## 3. Results and Discussion

### 3.1. ATR-FTIR of Monomers

Scheme 1 shows chemical formulae of terpene methacrylate monomers. Methacrylate monomers were obtained as colorless liquids with a yield of over 95% using the synthesis method described previously in [30,31,32,33]. The structure of monomers and their purity were confirmed by ^1^HNMR, ^13^CNMR, and FTIR and the data were previously presented in [34]. In this paper, only the ATR-FTIR spectra are shown (Figure 1) to show that compounds with the assumed structure were obtained. As can be seen, the process of esterification of geraniol or citronellol with methacryloyl chloride and the process of purifying a crude reaction product allow us to obtain terpene methacrylate monomers of high purity. As marked in Figure 1, absorption signals for all characteristic functional groups in the structure of methacrylate esters are visible. Moreover, basic properties of methacrylate monomers are presented in Table 2.

### 3.2. Conversion of Double Bonds

ATR-FTIR spectra for the selected copolymers post-cured at 120 °C are presented in Figure 2.

Additionally, the values of the degree of conversion (DC) are placed in Table 3. The degree of conversion of double bonds after irradiation of the samples was above 68%. A lower conversion of double bonds in the poly(citronellyl metacrylate) homopolymer indicates the lower reactivity of the citronellyl methacrylate monomer during UV polymerization as compared with the reactivity of the geranyl methacrylate monomer. For samples conditioned at 50 °C, an increase in DC to values above 80% was observed. The maximum conversion in the range of 90%–95% was obtained for the samples conditioned at 120 °C. The presented FTIR spectra for copolymers and the conversion studies proved that both types of double bonds (methacrylate and ethylenic double bonds) took part in the polymerization and post-curing processes. As a result, polymeric materials with cross-linked network structures were obtained (Scheme 2) [36,37].

### 3.3. Solubility Tests

Table 4 and Table 5 present the percentage change in mass in selected solvents for the tested polymeric materials conditioned at 50 °C (Table 2) and 120 °C (Table 3). By comparing the results in both tables, differences in the solubility of the polymeric materials conditioned at different temperatures can be noticed. The samples conditioned at 50 °C (series 1) are characterized by a lower solvent resistance as compared with the samples conditioned at 120 °C (series 2). The percentage change in mass loss is between 3.2% and 19.0% for the poly(geranyl methacrylate) homopolymer and between 2.1% and 17.6% for the poly(citronellyl methacrylate) homopolymer. The solubility of copolymers conditioned at 50 °C slightly increases with the increase in methyl methacrylate content, which may be due to the formation of less cross-linked structures and/or unreacted monomers. However, the solubility of all polymeric materials (homopolymers and copolymers) conditioned at 120 °C is very small. It is between 0% and 0.7% and it is independent of the content of methyl methacrylate in the copolymers. Such a low solubility of copolymers in polar and non-polar solvents may be caused by additional cross-linking reactions of carbon–carbon double bonds derived from aliphatic side chains (geranyl and citronellyl side chains) at higher temperatures. Moreover, comparing the results obtained for copolymers conditioned at both temperatures with the results obtained for the poly(methyl methacrylate) homopolymer, the copolymers are characterized by significantly higher solvent resistances in toluene and chloroform. However, the solvent resistances of copolymers conditioned at 120 °C in other solvents are comparable to that of poly(methyl methacrylate).

Additionally, the solubility of the copolymers conditioned at 50 °C depends on the type of solvent. Generally, the materials stand out by a higher solubility in non-polar solvents as compared with their solubility in polar solvents. This is predicted, since the formed polymers are hydrophobic-type materials that contain aliphatic substituents as side chains. The type of solvent has no effect on the solubility of the polymeric materials conditioned at 120 °C. This may be the result of additional cross-linking and the type of methacrylate monomers used for the polymerization process.

### 3.4. Chemical Resistance

As can be seen from Table 6 and Table 7, all tested polymers show high chemical stability in alkaline, acid, and buffer environments. After 6 months, the percentage of mass loss for polymeric materials conditioned at 50 °C was below 10%. However, the percentage of mass loss for polymers conditioned at 120 °C was below 0.5%. No mass change was observed for samples stored in 1M HCl and buffer with a pH of 5. The polymers conditioned at the higher temperature have higher chemical stability due to the formation of cross-links in the polymer network.

### 3.5. Glass Transition Temperature

The glass transition temperature (T_g_) for the polymeric materials conditioned at 120 °C was studied with the use of a DSC method. The course of DSC curves is presented in Figure 3. The T_g_ values are placed in Table 8. The Tg for poly(methyl methacrylate) obtained under experimental conditions is 91.7 °C. However, the poly(geranyl methacrylate) and poly(citronellyl methacrylate) homopolymers are characterized by much lower T_g_ values (24.1 °C and 3.1 °C, respectively). The T_g_ values for copolymers are directly depended on their composition. With increasing methyl methacrylate content in copolymers, an increase in T_g_ values is observed. Moreover, one T_g_ value is observed for all tested copolymers. This further confirms that the polymeric materials are copolymers and not blends.

### 3.6. Thermal Properties

Thermal properties of the obtained materials were studied with the use of a STA method. The course of the TG and DTG curves for polymeric materials conditioned at 50 °C (series 1) is presented in Figure 4. Moreover, thermal data, including the initial decomposition temperature (marked as the temperature where 5% of mass loss was observed (T_5%_)), the maximum decomposition temperature for each decomposition stage (T_max_), the mass loss in each decomposition stage (Δ_m_), and the residual mass at 540 °C (rm), are collected in Table 9. In addition, the thermal results for the tested materials conditioned at 120 °C (series 2) are placed in Table 10. As can be clearly seen, PMM conditioned at 50 °C is characterized by the highest thermal stability (218 °C). However, PGM and PCM show the following thermal stabilities: 158 °C and 153 °C, respectively. The thermal resistance of the copolymers depends on their content. The addition of MM to copolymers with GM and CM in each case leads to materials with higher thermal stability as compared with the PGM and PCM homopolymers. As the content of MM in the copolymers increases (to the value of 50 mass%), an increase in the thermal stability of the copolymers is observed. After exceeding the content of 50 mass% of MM, a decrease in the thermal stability of the polymeric materials is noticed.

Moreover, as can be seen from the data in Table 9, CM/MM copolymers are more thermally stable compared with GM/MM copolymers. Among the copolymers, copolymer 5 shows the highest thermal stability (224 °C).

The thermal analysis clearly confirmed that the tested materials decomposed in at least two main stages under inert conditions, with the note that the first decomposition stage may involve several undivided steps. The first decomposition stage is visible between T_5%_ and 320–340 °C with the T_max1_ given in Table 9. The second one appeared from temperatures of approx. 320–340 °C to 440–540 °C with the T_max2_ given in Table 9. Almost all tested materials decomposed completely (no residue) when heated to 540 °C (except PGM and copolymer 1).

However, the polymeric materials conditioned at 120 °C (series 2, Table 10) were characterized by a higher thermal resistance as compared with the samples conditioned at 50 °C. The PCM homopolymer was the most thermally stable material (251 °C). In turn, the PGM homopolymer had the lowest thermal stability (189 °C). The same relationship in thermal stability for copolymers conditioned at 50 °C and at 120 °C was noticed. With increasing content of MM in copolymers up to a value of 50 mass%, an increase in the value of T_5%_ was observed. In addition, except for copolymer 1, all of the polymeric materials showed thermal stability above 219 °C. Copolymers 1–6 decomposed in two well-marked stages. The first decomposition stage spreads from T_5%_ up to ca. 340 °C. The second one appeared between the temperatures of ca. 340 °C and 540 °C. All of the tested copolymers fully decomposed when heated to 540 °C (rm 0).

Comparing the T_5%_ values for series 1 and series 2, it can be concluded that the samples conditioned at 120 °C are more thermally stable. This indicates additional polymerization and cross-linking reactions in the tested materials at a higher temperature.

### 3.7. Simultaneous TG–FTIR Analysis

The decomposition course of the tested copolymers was analyzed using a coupled TG–FTIR method. The FTIR spectra of the emitted gaseous decomposition products gathered at T_max1_ and T_max2_ are presented in Figure 5 and Figure 6. As can be seen, on the FTIR spectra gathered at T_max1_ one can notice the presence of the following vibrations: the stretching vibrations for the OH group in water vapor at 3500–3900 cm^−1^; the stretching vibrations for =C–H at 3095 cm^−1^; the stretching vibrations for C–H at 2840–2980 cm^−1^; the stretching vibrations for C=O at 1733–1785 cm^−1^; the stretching vibrations for C=C at 1627–1633 cm^−1^ and 1670 cm^−1^; the deformation vibrations for C–C at 1375–1440 cm^−1^; the stretching vibrations for C–O at 1160–1300 cm^−1^; and the out-of-plane deformation vibrations characteristic of =C–H at 820–980 cm^−1^. The occurrence of the abovementioned vibrations characteristic of the functional groups of emitted volatiles may indicate the cleavage of C-C bonds in the structure of obtained copolymers. Thus, the release of geranyl methacrylate or citronellyl methacrylate esters as a result of the depolymerization of copolymers is clearly observed in the first decomposition stage. In addition, with respect to methacrylate ester emissions, the emission of water was also detected by the FTIR analysis. These observations are in accordance with our previous studies [29], where the emission of methacrylate esters as a result of the pyrolysis of other structure copolymers was indicated.

Meanwhile, in the second decomposition stage (T_max2_), the vibrations responsible for the characteristic groups in gaseous decomposition products at almost similar wavelengths appeared on the gaseous FTIR spectra. The absorption bands characteristic of the stretching vibrations for =C–H (3085 cm^−1^), the stretching vibrations for C–H (2842–2950 cm^−1^), the stretching vibrations for C=O (1735–1747 cm^−1^), the stretching vibrations for C=C (1633 cm^−1^), the deformation vibrations for C–H (1380–1445 cm^−1^), the stretching vibrations for C–O (1168–1300 cm^−1^), and the out-of-plane deformation vibrations for =C–H (811–990 cm^−1^) are clearly observed. The presence of these absorption bands confirms the formation of methyl methacrylate as the main decomposition product at the higher temperature (T_max2_). Besides the formation of methacrylate monomer, a small amount of water vapor (the bands above 3500 cm^−1^) is visible. The obtained results are in accordance with the literature data, where the depolymerization of poly(methyl methacrylate) is the main decomposition process [38,39].

## 4. Conclusions

The UV polymerization process was applied in order to prepare novel, more environmentally friendly polymeric materials from two methacrylate monomers: geranyl methacrylate or citronellyl methacrylate and methyl methacrylate. The performed tests proved that the copolymers conditioned at 120 °C had high solvent resistance and high chemical resistance, which were due to the formation of cross-links in the polymer network. The glass transition temperatures directly depended on the methyl methacrylate monomer (MM) content in the copolymers. An increase in T_g_ as the content of MM increased was observed. The thermal stability of the copolymers depended on the content of MM. It was in the range of 195–250 °C. The copolymers decomposed in at least two stages with the emission of methacrylate monomers as the main decomposition products.

In summary, the prepared polymeric materials, due to their properties, may find application as materials for the manufacture of parts operating in aggressive environments or at high temperatures, e.g., as column packing in gas or liquid chromatography or as machine parts in contact with solvents, acids, bases, buffers, coatings, etc.

## Data Availability

Not applicable.

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
