# Peer review of "UV Polymerization of Methacrylates—Preparation and Properties of Novel Copolymers"

_polymers, 2021, doi:10.3390/polym13101659_

Round 1
Reviewer 1 Report
Dear Editor
The current study describes the preparation and properties polymeric materials 8 manufactured during the UV polymerization process of methacrylate derivatives of terpene alcohols 9 (geranyl and citronellyl methacrylates) and methyl methacrylate. The overall novelty of the study is fine. I have few comments that need to be addressed before further considering the article.
Major English revision is needed. Technical writing is not up to the standards.
-Please re-write the abstract. Include the statement regarding the state of art and why this study was conducted.
-Also please briefly insert the methodologies used for conducting the study. (1 or 2 sentences).
-L74-77, Please re-write as it is confusing.
-L89-90, Please insert 'obtained' after 'were'.
Section 'Synthesis of Methacrylate Monomers', if the methodology is not new please give a reference.
-L179, what is meant by hesitat.
-Please discuss in detail the results in the solubility test with previous literature.
-Similarly, the authors need to discuss the thermal stability results and FT-IR results with previous literature studies. This discussion with help to improve the main purpose and novelty of the study.
-Insert major peaks in FT-IR figure.
-Conclusion is very long and results and values are given at large again. Please just give a brief conclusion discussing the main findings and how this study will contribute to future research of the subject.
-
Author Response
- Major English revision is needed. Technical writing is not up to the standards.
Answer: English was improved.
- Please re-write the abstract. Include the statement regarding the state of art and why this study was conducted.
Answer: Abstract was re-written. The aim of this study was added.
- Also please briefly insert the methodologies used for conducting the study. (1 or 2 sentences).
Answer: The methodologies were inserted more briefly.
- L74-77, Please re-write as it is confusing.
Answer: It was re-written.
- L89-90, Please insert 'obtained' after 'were'.
Answer: It was corrected.
- Section 'Synthesis of Methacrylate Monomers', if the methodology is not new please give a reference.
Answer: The methodology is not new. This methodology was described in my previous paper in Polymer for Advanced Technologies. Now, this description of methodology was removed. Suitable references were given [30-33].
- L179, what is meant by hesitat.
Answer: It was corrected.
- Please discuss in detail the results in the solubility test with previous literature.
Answer: The discussion of the solubility of the copolymers was extended. The results in the solubility of copolymers were compared with those obtained for homopolymers (PMM, PGM and PCM). Results for PMM were added to Tables. The obtained copolymers are novel polymeric materials, not described in the literature so far. On the other hand, there are a lot of copolymers and polymers with different structures. Therefore, it seems pointless to compare the properties of the copolymers with all known polymers. You can find the data in the tables for specific known polymers and compare them with those obtained in my work. This paper is not a review of articles so far published on polymers. This paper is a research paper that presents the receipt of new polymeric materials and their properties.
- Similarly, the authors need to discuss the thermal stability results and FT-IR results with previous literature studies. This discussion with help to improve the main purpose and novelty of the study.
Answer: As I have written above it seems pointless to compare the properties of the copolymers with all known polymers. The discussion the thermal stability results and FT-IR results were extended to include properties of PMM.
Main purpose and novelty of this study is to prepare new polymeric materials (copolymers) with the use of terpene methacrylate monomers and to evaluate their properties. This information was added to Abstract.
- Insert major peaks in FT-IR figure.
Answer: Major peaks in FT-IR figures (gaseous FTIR spectra) were inserted.
- Conclusion is very long and results and values are given at large again. Please just give a brief conclusion discussing the main findings and how this study will contribute to future research of the subject.
Answer: Conclusion section was improved as you suggested.
Reviewer 2 Report
The paper proposed by M. Worzakowska deals with the synthesis of methacrylate monomers derived from geraniol and citronellol and the characterization of solubility and thermal degradation of derived materials obtained by photopolymerization.
The paper requires major revisions. Indeed, a lot of information is missing.
- A state of art on the existing methods to synthesize methacrylate derivatives from terpene alcohols and/or on polymers resulting from such monomers should be performed to highlight the originality of this work.
- A scheme with chemical formulae of methacrylate monomers would be appreciated.
- Only FTIR spectra have been provided to attest the chemical structure of the synthesized methacrylate monomers. NMR characterization should be provided and some data, such as reaction yield, physical state … should be specified. Some of these data are already published by the author in Polym. Adv. Technol. 2018:29:1414-1425 but no reference to this paper was mentioned.
- There is no information on the photopolymerization step. What was the light intensity used to perform the photocuring? One hour of irradiation is extremely/abnormally high for such monomers. Was kinetic monitoring done to optimize this reaction time?
- The author mentions a thermal post-curing step without explanation. I deduce from the subsequent results that the conversion of methacrylate functions was not complete. So what is the ultimate conversion just after photocuring? And after post-curing at 50 and 120°C? Why were these two post-curing temperatures chosen? And considering the necessity of this post-polymerization step and the long time of irradiation under UV, what is the interest of using the UV process?
- Why are only solubility and thermal degradation of the photocured materials studied? For example, one would expect to know the Tg of the different photocured materials.
- The low solubility of photocured materials is attributed to additional crosslinking reactions involving ethylenic double bonds of geranyl and citronellyl chains. In what proportions are they involve in the crosslinking? A kinetic follow-up study by FTIR spectroscopy would be interesting to characterize this point more precisely. Moreover, should the author talk about copolymers in this paper since macromolecular chains are clearly not linear but crosslinked?
- There is no discussion of the influence of post-polymerization temperature on the solubility tests studied.
- PMMA should also be added as reference in the various results tables for comparison.
- The unit of absorbance in FTIR spectra cannot be in %. This does not make sense.
- What is the interest of figures 4 and 5 which show almost the same FTIR spectra?
Author Response
1.A state of art on the existing methods to synthesize methacrylate derivatives from terpene alcohols and/or on polymers resulting from such monomers should be performed to highlight the originality of this work.
Answer: The methodology of synthesis of terpene methacrylate monomers is not new. It was described in our previous papers in Ref. [30-33]. The preparation of polymers: homopolymers such as poly(geranyl methacrylate) and poly(citronellyl methacrylate) was described in my previous paper in Ref. 34. However the preparation of copolymers from these two terpene methacrylate monomers and conventionally available methyl methacrylate monomer and other conventionally available methacrylate and vinyl monomers has not been described in the literature so far. Only the preparation of copolymers from from pinene and styrene or pinene and acrylates is also known [21-25]. The polymerization of citronellol oxide obtained from citronellol for the production of hyperbranched polymers [26-28] or graft polymerization of methacrylate esters derivatives of citronellol, geraniol or nerol to the preparation of more environmentally friendly materials grafted on natural polymers with modified physicochemical properties have been described [29-32] is known. It was cited and described in the Introduction section.
The originality of this work is the preparation of novel copolymers, not described in the literature.
2.A scheme with chemical formulae of methacrylate monomers would be appreciated.
Answer: A scheme with chemical formulae of terpene methacrylate monomers was added as Scheme 1 in the manuscript.
3.Only FTIR spectra have been provided to attest the chemical structure of the synthesized methacrylate monomers. NMR characterization should be provided and some data, such as reaction yield, physical state … should be specified. Some of these data are already published by the author in Polym. Adv. Technol. 2018:29:1414-1425 but no reference to this paper was mentioned.
Answer: The characterization of terpene methacrylate monomers to confirm their structures (FTIR, 1HNMR, 13CNMR), reaction yield, physical state was presented in Polym. Adv. Technol. 2018:29:1414-1425. This reference was added as ref. 34. However, in Table 2, the basic properties of methacrylate monomers were placed.
4.There is no information on the photopolymerization step. What was the light intensity used to perform the photocuring? One hour of irradiation is extremely/abnormally high for such monomers. Was kinetic monitoring done to optimize this reaction time?
Answer: Information on the photopolymerization was added to section 2.4. Photopolymerization was initiated by ultraviolet light (340-365 nm). Photopolymerization was initiated with a TL20W/05 SLV low pressure mercury lamp.
One hour of irradiation – It was my typing mistake. It should be 10 min. Now it was corrected.
5.The author mentions a thermal post-curing step without explanation. I deduce from the subsequent results that the conversion of methacrylate functions was not complete. So what is the ultimate conversion just after photocuring? And after post-curing at 50 and 120°C? Why were these two post-curing temperatures chosen? And considering the necessity of this post-polymerization step and the long time of irradiation under UV, what is the interest of using the UV process?
Answer: Yes, you are right, after irradiation the conversion of double bonds was not complete. The post-curing temperatures were chosen based the FTIR studies and DSC studies of not complete cross-linked samples. In the manuscript, it was a typing mistake: it should be samples were conditioned at 50 ºC for 5 hours and at 120 ºC for 3 hours. However, our experimental studies confirmed that to obtain maximum conversion of double bonds (the same conversions as for longer time of conditioning), sample can be conditioned at 50ºC for 1 h and 120ºC for 1h. The post-curing temperatures were chosen experimentally. We use different post-curing temperatures and various post-curing time to optimize this process. The results of post-curing temperature and time were checked with the use of FTIR and DSC analysis. Samples were irradiated at 25ºC, the conversion was evaluated based on the FTIR, as it was described in Section 2.3.3. The degrees of conversion were placed in Table 3. Thus, in my studies the samples were post-cured in order to see if the conversion of double bonds increases at a higher temperature. In addition, use of higher cross-linking temperatures proved the polymerization of ethylenic double bonds from terpene methacrylate monomers. These double bonds do not polymerize in case of geraniol or citronellol under the same UV conditions. However, in terpene methacrylate esters such polymerization was observed.
As an example, the FTIR spectra for geranyl methacrylate monomer and poly(geranyl methacrylate) sample post-cured at 50ºC and 120ºC were placed below. However, the FTIR spectra for copolymers post-cured at 50ºC and 120ºC look similar to PGM. The spectra for copolymers were presented in the manuscript as Figure 2. As it is seen, some of double bonds do not take part in UV polymerization process, even in post-curing temperature of 50ºC. On the other hand, on DSC curves the exothermic cross-linking peak was visible for samples after irradiation and for samples post-cured at 50ºC. In turn, for samples post-cured at 120ºC, on the FTIR spectrum the signal responsible for the stretching vibrations for methacrylate bonds at 1633 cm-1 disappears. Moreover on the FTIR spectra for samples post-cured at 120ºC the intensity of the band characteristic for the stretching vibrations for ethylenic double bonds is very low which may indicate the reaction of these bonds and formation of cured polymeric materials. Similarly, on DSC curves no cross-linking peak was observed for samples post-cured at 120ºC.
The interest of these studies is to prepare novel materials with completely different properties than conventionally available poly(methyl methacrylate) and show that two types of bonds in terpene methacrylate monomers (methacrylate and ethylenic bonds) at a certain temperature can react in polymerization processes which lead to the preparation of cross-linked, branched polymeric materials.
6.Why are only solubility and thermal degradation of the photocured materials studied? For example, one would expect to know the Tg of the different photocured materials.
Answer: Tg was also studied with the use of DSC. The DSC curves were added as Figure 3. The Tg values were tabulated (Table 8).
We also plan to perform the mechanical properties, DMA, HDT, etc. However, now we have some technical problems to do cast of suitable dimensions. At this moment, we do not have the right forms to get flat, free from cracks and scratches bar for measurements.
7.The low solubility of photocured materials is attributed to additional crosslinking reactions involving ethylenic double bonds of geranyl and citronellyl chains. In what proportions are they involve in the crosslinking? A kinetic follow-up study by FTIR spectroscopy would be interesting to characterize this point more precisely. Moreover, should the author talk about copolymers in this paper since macromolecular chains are clearly not linear but crosslinked?
Answer: Yes, I agree that a kinetic follow-up study by FTIR spectroscopy would be interesting to characterize this point more precisely. Unfortunately we do not have the instrument which can measure on-line the FTIR spectra. We can irradiate the sample elsewhere and then transfer part of the sample to the ATR and measure the FTIR spectra. Additional crosslinking reactions involving ethylenic double bonds of geranyl and citronellyl chains was confirmed based on the FTIR spectra for the polymeric materials as it was presented in Fig. 3
Degree of conversion of double bonds was presented in Table 3.
I was wondering how to name these polymeric materials as they are crosslinked. I named them so because they were made of two monomers.
8.There is no discussion of the influence of post-polymerization temperature on the solubility tests studied.
Answer: The discussion was added.
9.PMMA should also be added as reference in the various results tables for comparison.
Answer: PMMA was added as reference material.
10.The unit of absorbance in FTIR spectra cannot be in %. This does not make sense.
Answer: The “%” was removed from FTIR spectra.
11.What is the interest of figures 4 and 5 which show almost the same FTIR spectra?
Answer: Figure 3 (now, Fig. 4) presents the gaseous FTIR spectra for geranyl copolymers. However, Figure 4 (now, Fig. 5) presents the gaseous FTIR spectra for citronellyl copolymers. The FTIR spectra were presented to show that under the decomposition of both types of copolymers, similar gases are released. If I did not show the FTIR spectra for both types of copolymers, the question might arise why these spectra were not included.

Reviewer 3 Report
The data collected and presented will be a contribution to the literature. However, additional information regarding the copolymer synthesis and the interpretation of results must be supplied. The measured changes in polymer properties (thermal stability, solubility, chemical resistance) with copolymer composition is explained by the hypothesis that the geranyl methacrylate (GM) and citronellyl methacrylate (CM) units undergo crosslinking upon conditioning of the copolymer films. However, neither the structures of the two monomers nor the crosslinking chemistry are presented. This information, as well as additional experimental details, needs to be added to the manuscript before publication.
Specific comments:
- Provide scheme/figure showing monomer structures, as they are not familiar to most readers.
- More details regarding the polymerization procedures are required in Section 2.3. What were the sample sizes and thicknesses? What was the UV source and intensity? How is it known that 100% conversion was achieved? The presence of residual monomer could influence the measured properties of the materials.
- While FTIR (Figure 1) shows the expected absorption signature of the monomers, it does not provide evidence regarding the purity of the compounds. Was NMR used to verify structure and purity of the GM and CM monomers?
- Section 3.2 describes how the samples conditioned at 50 C had lower solvent resistance compared to those at 120 C. Could this be due to incomplete monomer conversion? The author suggests that it is related to increased crosslinking reactions (line 179-182). The crosslinking mechanism should be presented as a scheme, and supporting references must be provided to support this interpretation.
- If crosslinking of the GM or CM groups occurs, wouldn’t there be a systematic trend in solubilities (Table 2) or chemical resistance (Table 4) with the copolymer composition? I would expect decreased cross-linking with increased MMA content in the copolymer, with the GM and CM homopolymers being the most highly crosslinked. However, this does not seem to be the case. Further discussion is warranted.
- The thermal stability of the copolymers may be related to the copolymer composition distribution, and how it changes as the polymerization goes from pure monomer to 100% conversion. Is anything known about the relative reactivity of GM and CM when copolymerized with MMA?
Author Response
The data collected and presented will be a contribution to the literature. However, additional information regarding the copolymer synthesis and the interpretation of results must be supplied. The measured changes in polymer properties (thermal stability, solubility, chemical resistance) with copolymer composition is explained by the hypothesis that the geranyl methacrylate (GM) and citronellyl methacrylate (CM) units undergo crosslinking upon conditioning of the copolymer films. However, neither the structures of the two monomers nor the crosslinking chemistry are presented. This information, as well as additional experimental details, needs to be added to the manuscript before publication.
Specific comments:
1.Provide scheme/figure showing monomer structures, as they are not familiar to most readers.
Answer: Structures of terpene methacrylate monomers were presented in Scheme 1.
2.More details regarding the polymerization procedures are required in Section 2.3. What were the sample sizes and thicknesses? What was the UV source and intensity? How is it known that 100% conversion was achieved? The presence of residual monomer could influence the measured properties of the materials.
Answer: More details are placed in Section 2.3.
The conversion was monitored by the FTIR for samples after irradiation, and post-curing at 50ºC and 120ºC. The conversions of double bonds were presented in Table 3.
Yes, the presence of unreacted monomer for copolymers conditioned at 50ºC may influence the measured properties. Such information was added in the text of the manuscript.
3.While FTIR (Figure 1) shows the expected absorption signature of the monomers, it does not provide evidence regarding the purity of the compounds. Was NMR used to verify structure and purity of the GM and CM monomers?
Answer: All the information regarding the characterization of terpene methacrylate monomers were presented in our previous paper in Ref. 34. This reference was added to the text of manuscript.
4.Section 3.2 describes how the samples conditioned at 50 C had lower solvent resistance compared to those at 120 C. Could this be due to incomplete monomer conversion? The author suggests that it is related to increased crosslinking reactions (line 179-182). The crosslinking mechanism should be presented as a scheme, and supporting references must be provided to support this interpretation.
Answer: Yes, it could be due to incomplete monomer conversion as it is proved based on the FTIR spectra. The part of the copolymer structure was presented in Scheme 2. The supporting references were added.
5.If crosslinking of the GM or CM groups occurs, wouldn’t there be a systematic trend in solubilities (Table 2) or chemical resistance (Table 4) with the copolymer composition? I would expect decreased cross-linking with increased MMA content in the copolymer, with the GM and CM homopolymers being the most highly crosslinked. However, this does not seem to be the case. Further discussion is warranted.
Answer: A systematic trend in solubilities and chemical resistance with the copolymer composition was observed. As the content of MM increased, slight increased in solubilities and chemical resistance of samples conditioned at 50ºC was observed. However for samples conditioned at 120ºC, MM has been incorporated into the cross-linked structure of the copolymers. Even the addition of 20% GM or CM to the copolymers resulted in their higher non-solubility and chemical resistance, despite the fact that the copolymer structure was less cross-linked. However, the content of GM and CM in copolymers influenced the glass transition temperature.
6.The thermal stability of the copolymers may be related to the copolymer composition distribution, and how it changes as the polymerization goes from pure monomer to 100% conversion. Is anything known about the relative reactivity of GM and CM when copolymerized with MMA?
Answer: The thermal stability of the copolymers was related to the copolymer composition as it was described in the text of the manuscript. The thermal stability of pure monomer cannot be determined as it polymerizes under the influence of temperature into the corresponding polymer. Thus, during such studies the thermal stability of polymer, not the monomer, can be determined.
The relative reactivity of GM was higher than the relative reactivity of CM. After 10 minutes of irradiation, the degree of double bonds conversion in GM copolymers was higher than for CM copolymers.
Round 2
Reviewer 1 Report
It can be accepted now.
Reviewer 2 Report
The second version of M. Worzakowska’s paper has been corrected overall as suggested. Nevertheless, the additional information requested and provided leads to new questions or remarks.
In Abstract: “in aggressive conditions” would be more appropriate than “in corrosive conditions”
In Part 2. Materials and methods
- Part 2.3.: The author says that samples were irradiated for 10 min. The light intensity used to perform the photocuring is always not specified. Was this irradiation time optimized?
- Part 2.3.3.: On which side of the sample were the FTIR spectra measured? As mentioned in section 2.3.3. the photo-crosslinked samples are thick (5 mm) and there is probably a conversion gradient between the side of the sample exposed to UV radiation and the other side.
- Part 2.3.4.: in this part and in the whole document, talking about swelling and extractables rate would be more appropriate than solubility since the copolymers are crosslinked.
- Table 1: “Composition of studied homo- and co-polymers” would be a more appropriate title.
- 2.3. is really § 2.4. and so on
In Part 3. Results and discussion
- Part 3.1.: I don't agree with the expression "as can be seen”; indeed, Figure 1 only shows FTIR spectra of final products, which Is not sufficient to judge their purity. The FTIR spectra of geraniol and citronellol should also be presented.
- Figures 1 and 2 always show the absorbance unit in %. This must be corrected.
- Figure 2: Rather than showing the FTIR spectra of the three copolymers derived from geranyl methacrylate which are very similar, I would find it more interesting to show the FTIR spectra of one of these copolymers before and after UV curing.
- I don’t think that Scheme 2 is really useful.
- Part 3.5.: The determination of the Tg of the copolymers by DSC seems very hazardous for some of them. For example, the DSC curves of copolymers 2 and 3 seem almost superposable and yet, the author gives Tg values with a 20°C difference. A measurement by DMA would perhaps be more precise.
Reviewer 3 Report
The additional experimental data supplied improves the manuscript significantly. All issues have been addressed, and I recommend acceptance without further change.